# Isolating Latent Structure with Cross-population Variational Autoencoders

## Abstract

A significant body of recent work has examined variational autoencoders as a powerful approach for tasks which involve modeling the distribution of complex data such as images and text. In this work, we present a framework for modeling multiple data sets which come from differing distributions but which share some common latent structure. By incorporating architectural constraints and using a mutual information regularized form of the variational objective, our method successfully models differing data populations while explicitly encouraging the isolation of the shared and private latent factors. This enables our model to learn useful shared structure across similar tasks and to disentangle cross-population representations in a weakly supervised way. We demonstrate the utility of our method on several applications including image denoising, sub-group discovery, and continual learning.

## 1 Introduction

Unsupervised learning of latent representations is widely used for dimensionality reduction, density estimation, and structure or sub-group discovery among other applications. Methods for recovering such representations typically rely on the assumption that the observed data is a manifestation of only a limited number of factors of variation (Locatello et al., 2019; Bengio et al., 2013). The variational autoencoder (VAE) (Kingma & Welling, 2013), a combination of a non-linear latent variable model and an amortized inference scheme (Dayan et al., 1995), is a popular method for recovering such latent structure. VAEs and their extensions have received considerable attention in recent years and have been shown useful for modeling text (Miao et al., 2016), images (Gulrajani et al., 2016), and other data exhibiting complex correlations (Gómez-Bombarelli et al., 2018). However, barring a few exceptions (Bouchacourt et al., 2018; Severson et al., 2019), this line of work has assumed the data to be independent and identically distributed.

In this work, we consider the task of modeling independent but not identically distributed data. We are particularly interested in analyzing data comprising two or more distinct but related sub-populations. Such data arise frequently in practice. For instance, patients suffering from an ailment may exhibit symptomatic heterogeneity based on gender or environmental factors, documents in a corpus may exhibit semantic or syntactic similarities based on genre or authorship, images may cluster depending on the image subject. Our goal is to provide rich descriptions of such data by recovering latent representations that disentangle factors of variation common to all populations from those that are unique to a particular population. Motivated by this challenge, we propose non-linear latent variable models that explicitly account for the heterogeneity in the data. Coupled with an inference procedure that encourages high (low) mutual information between the latent representation and the data within (across) a population, we show that our models are indeed able to recover population-specific representations that are salient and disentangled across differing populations as well as shared representations that isolate commonalities between populations.

Through careful experiments, we vet the effectiveness of our approach and demonstrate that the learned representations are useful for a diverse set of applications including image denoising, unsupervised sub-group discovery, and continual learning.

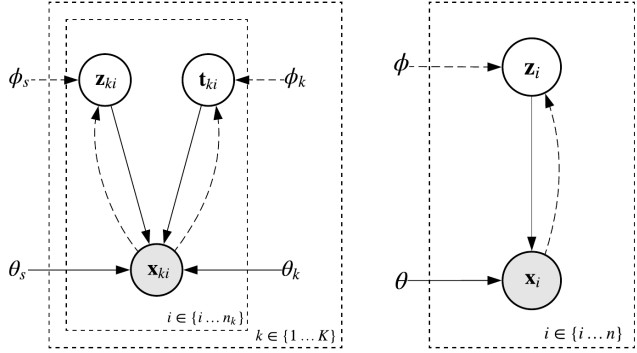

Figure 1: Graphical model of CPVAE (left) vs. standard VAE (right)

## 2 METHOD

### 2.1 CROSS-POPULATION VARIATIONAL AUTOENCODER

We propose a generative model of a data instance $\mathbf{x}_{ki}$ belonging to population $k$ using two latent variables $\mathbf{z}_{ki}$ and $\mathbf{t}_{ki}$. The latent variables are mapped to the observed space using non-linear mappings, $f_{\theta_s}(\mathbf{z}_{ki})$ and $f_{\theta_k}(\mathbf{t}_{ki})$, each parameterized by a neural network. While we share the parameters $\theta_s$ among all populations, $\theta_k$ are only shared among instances belonging to the population $k$. This construction encourages the model to capture common latent structure in $\theta_s$, while allowing $\theta_k$ to focus on factors of variation unique to the particular population $k$. We combine the contributions from the two mappings using a function $g$. The generative procedure can be summarized as,

$$
\begin{aligned}
\mathbf{z}_{ki} &\sim \mathcal{N}(0, \mathbf{I}), \qquad \mathbf{t}_{ki} \sim \mathcal{N}(0, \mathbf{I}), \\
\mathbf{x}_{ki} \mid \mathbf{z}_{ki}, \mathbf{t}_{ki} &\sim p(g(f_{\theta_s}(\mathbf{z}_i), f_{\theta_k}(\mathbf{t}_{ki}))), \quad \forall i \in \{1 \ldots n_k\}, \forall k \in \{1, \ldots, K\},
\end{aligned}
\tag{1}
$$

where $p$ is an appropriately chosen distribution for modeling the observed data. In our experiments, we use an additive function $g(a, b) = a + b$, and select $p$ to be the Gaussian distribution with mean parameterized by $g$ and an isotropic diagonal covariance matrix, $\Psi$. We emphasize that other choices of function can easily be incorporated into the model. Exploring the space of aggregation functions is planned future work.

### 2.2 AMORTIZED VARIATIONAL INFERENCE

Amoritized variational inference (Dayan et al., 1995; Gershman & Goodman, 2014) with the aid of reparameterized gradients (Rezende et al., 2014; Kingma & Welling, 2013) is straightforward to implement for the model described in equation 1. We assume that the variational approximation factorizes conditioned on the observation,

$$
q_\phi(\mathbf{z}_{ki}, \mathbf{t}_{ki} \mid \mathbf{x}_{ki}) = q_{\phi_s}(\mathbf{z}_{ki} \mid \mathbf{x}_{ki}) q_{\phi_k}(\mathbf{t}_{ki} \mid \mathbf{x}_{ki}).
\tag{2}
$$

We parameterize the variational distribution with a single, shared inference network with two distinct outputs, one for each latent variable. Finally, the model and variational parameters are jointly learned by optimizing the evidence lower bound (ELBO),

$$
\begin{aligned}
\mathcal{L}(\theta, \phi) = \sum_{k=1}^{K} \sum_{i=1}^{n_k} &\mathbb{E}_{q_\phi(\mathbf{z}_{ki}, \mathbf{t}_{ki} \mid \mathbf{x}_{ki})} [\log p(\mathbf{x}_{ki} \mid \mathbf{z}_{ki}, \mathbf{t}_{ki}; \theta_s, \theta_k)] \\
&- D_{\mathrm{KL}}\left(q_\phi(\mathbf{z}_{ki}, \mathbf{t}_{ki} \mid \mathbf{x}_{ki}) || p(\mathbf{z}_{ki}, \mathbf{t}_{ki})\right).
\end{aligned}
\tag{3}
$$

Importantly, in the above setup, the per-population model and variational parameters $\theta_k$ and $\phi_k$ are learned only from the data in that population, allowing them to learn population specific representations. We refer to this combination of the model described in equation 1 and the inference network defined above as the cross-population variational auto-encoder (CPVAE), owing to its similarity with the VAE (Kingma & Welling, 2013). See Figure 1 for a graphical representation of the CPVAE and comparison to the VAE.

## 2.3 MUTUAL INFORMATION REGULARIZED INFERENCE

By employing population-specific mappings, CPVAE prevents latent factors of variation unique to one population from 'leaking' into the private representation of another population. However, the model structure does not prevent leakage between the shared and population specific representations, since every data instance is generated by a combination of the shared and population specific representation. In fact, when CPVAE is trained by maximizing the ELBO in equation 3 we find that the private representations of a population often exhibit latent features from the shared space and vice versa.

Inspired in part by the InfoVAE (Zhao et al., 2017), we discourage such inferences by minimizing the mutual information between population-specific latent variables and data from non-corresponding populations while maximizing the mutual information within a population.

Let $\mathbf{x}_k = \{\mathbf{x}_{k1}, \ldots, \mathbf{x}_{kn_k}\}$ denote the set of all data instances belonging to population $k$, let $\mathbf{t}_k, \mathbf{z}_k$ be the set of corresponding latent variables, and $\mathbf{x}_{-k} = \{\mathbf{x}_j; \forall j \neq k \in K\}$ denote the set of all *non-corresponding* populations. We learn the CPVAE by maximizing,

$$J(\theta, \phi) = \mathcal{L}(\theta, \phi) + \sum_{k}^{K} n_k \left( I_q(\mathbf{x}_k; \mathbf{t}_k) - I_q(\mathbf{x}_{-k}; \tilde{\mathbf{t}}_k) \right), \tag{4}$$

where $\tilde{\mathbf{t}}_k = \{\{\tilde{\mathbf{t}}_{ji}\}_{i=1}^{n_j}\}_{j \neq k}$ and $\tilde{\mathbf{t}}_{ji} \sim q_{\phi_k}(\tilde{\mathbf{t}}_{ji} \mid \mathbf{x}_{ji})$ where $\mathbf{x}_{ji} \in \mathbf{x}_{-k}$, i.e., the result of encoding members of the non-corresponding populations, $\mathbf{x}_{-k}$ with the parameters $\phi_k$ of the current population's inference network. This function can be simplified to a tractable objective by re-writing the mutual information term:

$$
\begin{aligned}
I_q(\mathbf{x}_k; \mathbf{t}_k) &= - \mathbb{E}_{q_{\phi_k}(\mathbf{x}_k, \mathbf{t}_k)} \log \frac{q_{\phi_k}(\mathbf{t}_k)}{q_{\phi_k}(\mathbf{t}_k|\mathbf{x}_k)} \\
&= - \mathbb{E}_{q_{\phi_k}(\mathbf{x}_k, \mathbf{t}_k)} \log \frac{q_{\phi_k}(\mathbf{t}_k)p(\mathbf{t}_k)}{q_{\phi_k}(\mathbf{t}_k|\mathbf{x}_k)p(\mathbf{t}_k)} \\
&= - \mathbb{E}_{p_D(\mathbf{x}_k)}\mathbb{E}_{q_{\phi_k}(\mathbf{t}_k|\mathbf{x}_k)} \log \frac{p(\mathbf{t}_k)}{q_{\phi_k}(\mathbf{t}_k|\mathbf{x}_k)} - \mathbb{E}_{q_{\phi_k}(\mathbf{t}_k)} \log \frac{q_{\phi_k}(\mathbf{t}_k)}{p(\mathbf{t}_k)} \\
&= \frac{1}{n_k} \sum_{i=1}^{n_k} [D_{\mathrm{KL}}\left(q_{\phi_k}(\mathbf{t}_{ki}|\mathbf{x}_{ki})||p(\mathbf{t}_{ki})\right)] - D_{\mathrm{KL}}\left(q_{\phi_k}(\mathbf{t}_k)||p(\mathbf{t}_k)\right),
\end{aligned}
\tag{5}
$$

where we assume $q_{\phi_k}(\mathbf{x}_k, \mathbf{t}_k) = p_D(\mathbf{x}_k)q_{\phi_k}(\mathbf{t}_k \mid \mathbf{x}_k)$ and $p_D(\mathbf{x}_k)$ is the empirical distribution. Note that when we take $I_q(\mathbf{x}^k; \mathbf{t}^k) - I_q(\mathbf{x}^{-k}; \mathbf{t}^k)$ as in our objective, the intractable marginal KL term conveniently cancels out. The complete objective function is

$$
\begin{aligned}
J(\theta, \phi) = \sum_{k=1}^{K} \Bigg[ &\sum_{i=1}^{n_k} \left[ \mathbb{E}_{q_\phi(\mathbf{z}_{ki}, \mathbf{t}_{ki}|\mathbf{x}_{ki})}[\log p(\mathbf{x}_{ki} \mid \mathbf{z}_{ki}, \mathbf{t}_{ki}; \theta_s, \theta_k)] - D_{\mathrm{KL}}\left(q_{\phi_s}(\mathbf{z}_{ki}|\mathbf{x}_{ki})||p(\mathbf{z}_{ki})\right) \right] \\
&- \frac{n_k}{K-1} \sum_{j \neq k}^{K} \frac{1}{n_j} \sum_{i=1}^{n_j} D_{\mathrm{KL}}\left(q_{\phi_k}\left(\tilde{\mathbf{t}}_{ji}|\mathbf{x}_{ji}\right)||p(\mathbf{t}_{ji})\right) \Bigg],
\end{aligned}
\tag{6}
$$

This objective differs from the standard variational objective only in that the divergence of the private latent prior for any population $\mathbf{t}_k$ is minimized for its contrasting populations $\mathbf{x}_{-k}$ and not its corresponding population $\mathbf{x}_k$. That is, we minimize $D_{\mathrm{KL}}\left(q_{\phi_k}(\tilde{\mathbf{t}}_k|\mathbf{x}_{-k})||p(\mathbf{t}_k)\right)$ rather than $D_{\mathrm{KL}}\left(q_{\phi_k}(\mathbf{t}_k|\mathbf{x}_k)||p(\mathbf{t}_k)\right)$. As we will show, this modification reduces entanglement between private latent vectors and enables the model to learn more salient representations of each population. Moreover, since Kullback Leibler divergence is always non-negative, $J(\theta, \phi)$ in equation 6 remains a lower bound to the marginal likelihood.

In some scenarios, it may desirable to increase the importance of the cross mutual information term $I_q(\mathbf{x}_{-k}; \mathbf{t}_k)$ to our objective, further discouraging shared features from leaking into population-specific representations. To this end, we can simply add a scaling constant $\alpha \geq 1$ to the final term in equation 6. In our experiments, we often find substantially improved results by beginning

with $\alpha = 1$ and gradually annealing its value over training. A summary of the complete training procedure can be found in Algorithm 1.

---

**Algorithm 1** Training procedure of CPVAE

---

Initialize conditional parameters $\theta = \{\theta_s\} \cup \{\theta_k; \forall k \in K\}$;
Initialize variational parameters $\phi = \{\phi_s\} \cup \{\phi_k; \forall k \in K\}$;
**repeat**
    Sample mini-batch from each population $\{\{\mathbf{x}_{ki}\}_{i=1}^{m_k}\}_{k=1}^{K}$
    **for** $k \in 1 \dots K$ **do**
        Sample shared codes $\mathbf{z}_k \sim q_{\phi_s}(\mathbf{z}_k|\mathbf{x}_k)$;
        Sample private codes $\mathbf{t}_k \sim q_{\phi_k}(\mathbf{t}_k|\mathbf{x}_k)$;
        **for** $j \neq k \in 1 \dots K$ **do**
            Sample fictitious codes $\tilde{\mathbf{t}}_j \sim q_{\phi_k}(\tilde{\mathbf{t}}_j|\mathbf{x}_j)$;
        **end**
    **end**
    Calculate $J(\theta, \phi)$ as in (6);
    Update $\theta^{t+1}, \phi^{t+1} \leftarrow \theta^t, \phi^t$ according to ascending gradient estimate of $J(\theta, \phi)$;
**until** *convergence*;

---

## 3 RELATED WORK

Substantial recent work has explored methods for the unsupervised learning of *disentangled* representations. Though lacking a formal definition, the key idea behind a disentangled representation is that it should separate distinct informative factors of variation of the data (Bengio et al., 2013; Locatello et al., 2019). Several techniques have been proposed to encourage VAEs to learn disentangled latent representations. Some examples are $\beta$-VAE (Higgins et al., 2017), AnnealedVAE (Burgess et al., 2018), FactorVAE (Kim & Mnih, 2018), $\beta$-TCVAE (Chen et al., 2018), and DIP-VAE (Kumar et al., 2017), each of which proposes some variant of the VAE objective to encourage the variational distribution to be factorizble. Recently, it has been proposed that it is not possible to recover disentangled features without inductive bias or supervision (Locatello et al., 2019). CPVAE makes no assumptions about the composition of variational factors but imposes a form of weak supervision where population assignment is known and uses that information in choosing the model architecture and learning algorithm.

A few other techniques have been proposed to use weak supervision for learning improved representations. Multi-study factor analysis (De Vito et al., 2018a;b) has similar aims and structure as compared to CPVAE but uses a linear model and focuses on applications related to high-throughput biological assay data. Contrastive latent variable models (Severson et al., 2019; Abid & Zou, 2019) have non-linear variants but focus on the case where one dataset is the target to be compared/contrastive to another dataset. Multi-level variational autoencoders (ML-VAEs) have been proposed as a way to incorporate group-level data in unsupervised learning (Bouchacourt et al., 2018). After dividing data into disjoint groups according to some factor of interest, the ML-VAE framework models latent structure both at the level of individual observations and of entire groups. This method effectively separates latent representations into semantically relevant parts, but differs from the CPVAE framework which models both private and shared structure at the observation level. Output-interpretable VAEs (oi-VAEs) have been proposed to leverage data that can be partitioned into within sample groups in an interpretable model (Ainsworth et al., 2018). The model structure is such that the components within each group are modeled with separate generative networks. Mappings from the latent representations to each of the groups are encouraged to be sparse via hierarchical Bayesian priors to improve interpretability. The primary difference between oi-VAE and CPVAE is that oi-VAE uses groupings over components, $\mathbf{x}_i \in \mathbb{R}^d$, $\mathbf{x}_i = [\mathbf{x}_{i1}, \dots, \mathbf{x}_{iK}]$, and $\mathcal{D} = \{\mathbf{x}_i\}_{i=1}^{n}$ and CPVAE uses groupings over instances, $\mathbf{x}_{ki} \in \mathbb{R}^d$ and $\mathcal{D} = \{\{\mathbf{x}_{ki}\}_{i=1}^{n_k}\}_{k=1}^{K}$. There has also been recent work in learning disentangled representations from sequential data (Hsu et al., 2017; Li & Mandt, 2018). These models share a representation across all elements of the sequence to learn global sequence dynamics while local aspects are modeled via time step specific representations. This problem is somewhere between the standard disentangled representation learning, where the goal is to learn disentangled features with no prior knowledge, and weak supervision as the sequential nature of the

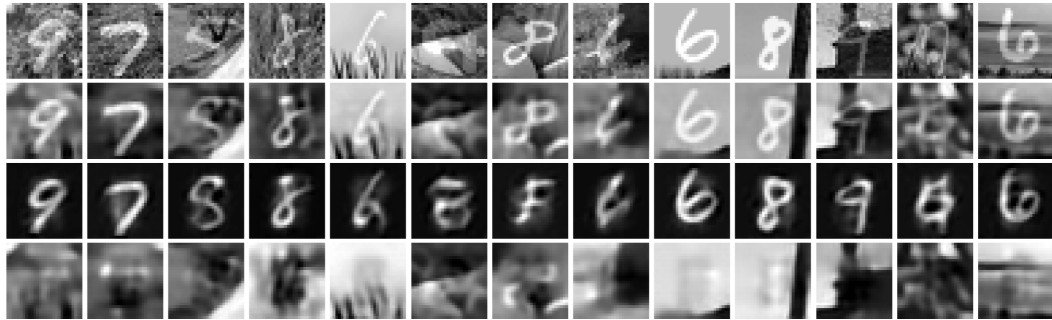

Figure 2: Grassy-MNIST reconstructions with two populations: digits 0-4 and digits 5-9. First row: original images. Second row: full reconstructions. Third row: private space (digit-only) reconstructions. Fourth row: shared space (background only) reconstructions. Note that this model has never seen the original digits or the original backgrounds.

| Method | SHARED ARI | PRIVATE ARI |
|---|---|---|
| VAE | .1881 | — |
| CPVAE, NO MI | .0306 | .1926 |
| CPVAE, $\alpha = 1$ | .0031 | .2281 |

Table 1: Adjusted rand index (ARI) for discovered clusters in the shared vs. private latent spaces.

data inherently provides some structure. Our work presented here focuses on non-sequential data. Moreover, unlike us these works do not employ mutual information regularized inference, which we find to be crucial for recovering disentangled shared and private representations.

## 4 EXPERIMENTS

In order to determine the effectiveness of our approach as well as demonstrate several possible applications, we evaluate it with a number of experiments on tasks including image denoising, subgroup discovery, classification, and continual learning.

Each experiment employs the following setup: the encoder and decoder are strided convolutional neural networks with ReLU activations and batch normalization after each layer. The decoder employs residual skip connections which have been shown to help prevent posterior collapse in VAEs (Dieng et al., 2018). We optimize using Adam (Kingma & Ba, 2014) with a base learning rate of 0.001.

### 4.1 DENOISED GENERATIVE MODELING APPLIED TO GRASSY-MNIST

We evaluate our model on a synthetic dataset of handwritten digits from the MNIST (LeCun et al., 1998) superimposed on grassy backgrounds from ImageNet (Russakovsky et al., 2015) (see Figure 2, top row for example images). In Abid et al. (2018); Severson et al. (2019); Abid & Zou (2019), the authors train contrastive models on this dataset along with the original grass images in order to learn more salient latent representations of the digits.

In our experiment, we split this synthetic data into two populations consisting of digits 0-4 and digits 5-9. We use a shared latent dimension of $100$ and a population-specific latent size of $25$. We show that our model is able to effectively separate the complex background representations in the shared latent space from that of the digits in each private space. Crucially, the model is never shown either the original background grass images nor the original non-noisy digits.

A sample of reconstructed images can be seen in Figure 2. The first and second rows show the original digits with noise and the full CPVAE reconstructions, respectively. The third and fourth rows show the reconstructions when the shared or the private latent spaces are ignored, respectively.

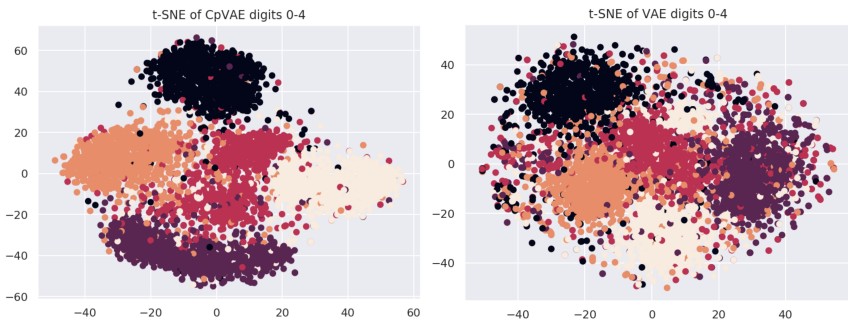

Figure 3: Visualization of subgroups within MNIST digit population after t-SNE projection of latent space for our method (left) vs a standard VAE (right). Each color represents one of the five digits within the population.

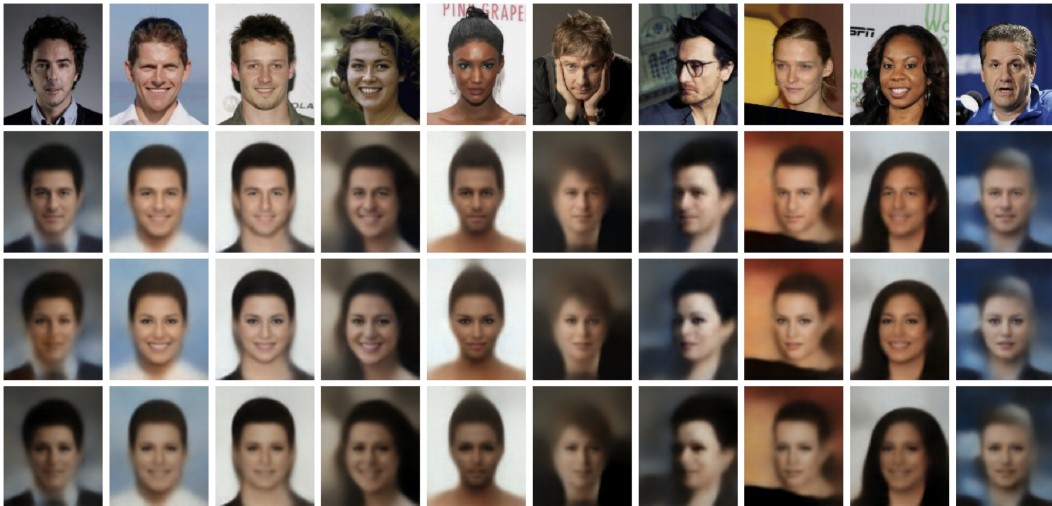

Figure 4: CPVAE reconstructions of CelebA dataset images. Top row: original images. Second row: reconstructions with `male` and shared space decoders. Third row: reconstructions with `non-male` and shared space decoders. Bottom row: shared decoder only.

These results qualitatively demonstrate CPVAE's ability to separate the shared and private feature representations.

As an evaluation of the salience of the private space representations, we also measure the ability of our model to do sub-group discovery within each population. K-means clustering is applied to the private space latent representations and compared to the true class labels using the adjusted rand index (ARI), which measures the correspondence of the cluster assignments with the true labels (Menezes & Roth, 2017). The results are in Table 1 and a TSNE visualization of one of the private latent spaces for this task compared with that of a standard VAE is in Figure 3. Our model outperforms the standard VAE, demonstrating the ability of our model to learn improved representations by incorporating cross-population structure. Incorporating the MI objective ($\alpha = 1$) further improves ARI scores in the private space and worsens scores in the shared space, suggesting that the MI term effectively mitigates the leakage of population-specific features in the shared space.

## 4.2 DISENTANGLING LABELED ATTRIBUTES IN CELEBRITY IMAGES

CPVAE allows us to model data such that latent structure which corresponds to some labeled attribute of interest can be isolated. To demonstrate this, we perform an experiment on the Large-scale CelebFaces Attributes (CelebA) dataset (Liu et al., 2015). This data consists of 202,599 aligned and cropped pictures of celebrities with 40 binary attributes labeled for each image. See Figure 4.

| DATASET | ANNEAL $\alpha$ | $\alpha = 1$ | NO MI | RESNET50 |
|---|---|---|---|---|
| MNIST | 99.5 | 98.2 | 97.3 | 99.6 |
| MNIST-GRASS | 77.5 | 75.1 | 70.5 | 85.7 |
| CIFAR-10 | 76.2 | 47.9 | 42.2 | 84.1 |

Table 2: Maximum likelihood classification test accuracies (%) for our model with and without mutual information terms evaluated on different datasets. For reference, we also include accuracies from a ResNet50 classifier.

We train a CPVAE model on this dataset with two populations determined by the `male` attribute label. Under this setup, the model is incentivized to learn representations of gender-specific features in the private latent spaces while the shared space infers the remaining factors of variation. As a qualitative evaluation of this model, we autoencode a sample of images with each private space and examine the resulting reconstructions. By reconstructing an image with the non-corresponding population's decoder, we get reconstructions that closely resemble the original image but with features that appear traditionally male when constructed from the male space and female when constructed from the non-male space. See Figure 4 for a sample of these results. This serves as additional evidence of our model's ability to separate population-specific features from shared latent structure.

### 4.3 MAXIMUM MARGINAL LIKELIHOOD CLASSIFICATION

As an additional evaluation of our model's ability to learn disentangled population-specific representations, we test our model's ability to classify unseen data points into their corresponding populations. We assign an instance $\mathbf{x}_*$ to the population that maximizes its marginal likelihood,

$$\hat{k}_i = \arg\max_{k \in K} p(\mathbf{x}_* \mid \theta_k, \theta_s) = \arg\max_{k \in K} \mathbb{E}_{p(\mathbf{z}_{ki}, \mathbf{t}_{ki})} \left[ p(\mathbf{x}_* \mid \mathbf{z}_{ki}, \mathbf{t}_{ki}; \theta_k, \theta_s) \right]. \tag{7}$$

We use importance sampling to compute the intractable expectation,

$$\mathbb{E}_{p(\mathbf{z}_{ki}, \mathbf{t}_{ki})} \left[ p(\mathbf{x}_* \mid \mathbf{z}_{ki}, \mathbf{t}_{ki}; \theta_k, \theta_s) \right] = \mathbb{E}_{q_\phi(\mathbf{z}_{ki}, \mathbf{t}_{ki} \mid \mathbf{x}_*)} \left[ p(\mathbf{x}_* \mid \mathbf{z}_{ki}, \mathbf{t}_{ki}; \theta_k, \theta_s) \frac{p(\mathbf{z}_{ki}, \mathbf{t}_{ki})}{q_\phi(\mathbf{z}_{ki}, \mathbf{t}_{ki} \mid \mathbf{x}_*)} \right]. \tag{8}$$

In order to achieve a high accuracy, the model must learn features in each population-specific latent space which are unique to its corresponding set. We therefore evaluate our model's classification performance on several labeled image datasets of varying difficulty — MNIST, CIFAR-10 (Krizhevsky et al., 2009), and the Grassy MNIST described in experiments above. In each case, we define a distinct population for each class and evaluate its performance on a held-out test set. We emphasize that our goal with this experiment is not to demonstrate state-of-the-art classification performance, but to provide a convenient, quantitative benchmark for evaluating the quality of the model's learned representations. For reference, we also provide classification accuracies from a ResNet-50 convolutional model (He et al., 2015; LeCun et al., 2015) trained by maximizing $p(k_* \mid \mathbf{x}_*)$ for five hundred epochs. Note that these CNN scores are not the state of the art for each task, but serve as a contextual reference point for understanding our model's performance. We compare this against our CPVAE models both with and without the mutual information regularization as well as a variant where the weight on the mutual information term, $\alpha$, is gradually annealed over training. Additional details about model architectures and the $\alpha$ annealing scheme can be found in the supplement.

The results can be seen in Table 2. Our model approaches the performance of the convolutional neural network despite not being trained to directly maximize classification performance. We find that performance improves by utilizing our mutual information regularized objective, particularly when $\alpha$ is annealed over training. This result provides compelling evidence that our model is able to effectively learn private space representations which are unique to each corresponding population.

### 4.4 CONTINUAL LEARNING

Neural networks are widely known to be susceptible to a phenomenon known as *catastrophic forgetting* (McCloskey & Cohen, 1989), wherein a model's performance on a previously learned task

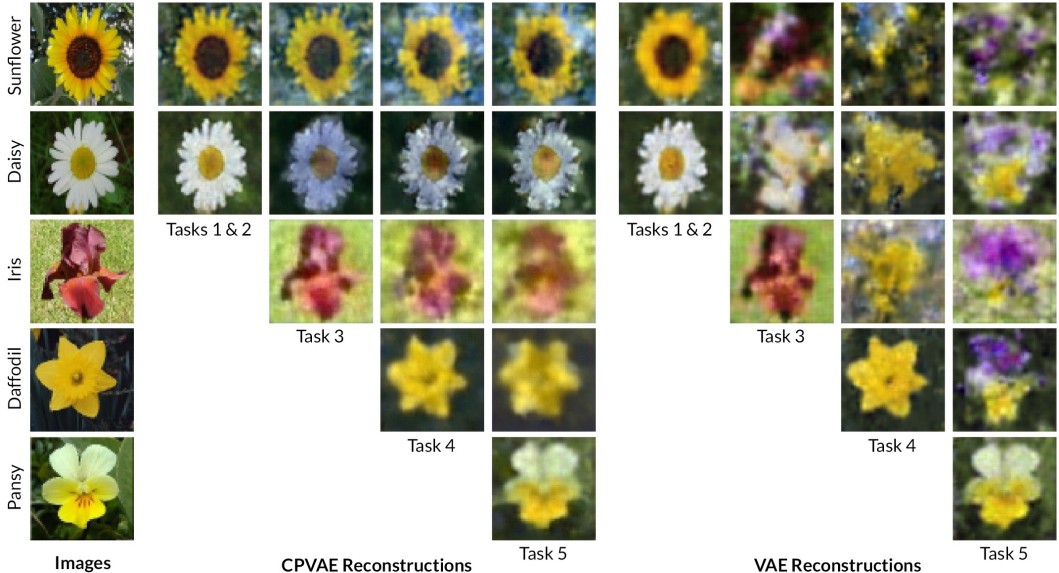

Figure 5: Image reconstructions for CPVAE vs VAE in continual learning setting. Each column shows the reconstructions for the labeled task as well as previous tasks, demonstrating CPVAE's ability to retain previously learned information by utilizing both population-specific and shared latent vectors.

degrades rapidly as new tasks are encountered. We show the merit of our method in mitigating this phenomenon in the context of variational autotencoders. By modeling sequentially arriving tasks as distinct populations, our method models the structure unique to each task in the population-specific latent spaces and learns general information useful to all tasks in the shared space.

We perform an experiment on the 17 Category Flower dataset (Nilsback & Zisserman, 2006). We evaluate our model's ability to reconstruct images from five sequentially-arriving image categories: *sunflower*, *daisy*, *iris*, *daffodil*, and *pansy*. Each category is assigned its own population and training proceeds in two phases: first, tasks 1 and 2 are learned together using the mutual information regularized objective. This step allows the model to infer structure that is shared between the two tasks that can then be utilized for learning later tasks. For example, the shared space may learn to represent the backgrounds while the private spaces learn to represent the corresponding flowers. Next, each remaining task is learned in sequence with the evidence lower bound.

The results can be seen in Figure 5 where we show the reconstructions for an image from each task throughout the training procedure for CPVAE in comparison to a VAE. In contrast to the VAE whose reconstructions are profoundly impacted by introduction of new tasks, the reconstructions from our model are relatively unaffected.

## 5 CONCLUSION

In this work, we presented a framework for using a VAE-like architecture to model multiple sets of data which are independent but come from differing distributions. We developed an architecture which encourages the isolation of shared and private latent factors, and presented a mutual information regularized version of the evidence lower bound which discourages entanglement of the shared and population-specific latent vectors. Our experiments on the Grassy MNIST dataset demonstrated our model's ability to learn more salient representations and to effectively separate the shared and private latent factors on the task of image denoising. We also showed the effectiveness of our regularized objective in learning population-specific representations on several image classification tasks. Lastly, we demonstrated the value of learning population-specific representations in the context of continual learning where our method retains performance on previously encountered tasks in comparison to a VAE.

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

# Appendices

## A GRASSY MNIST GENERATIVE SAMPLES

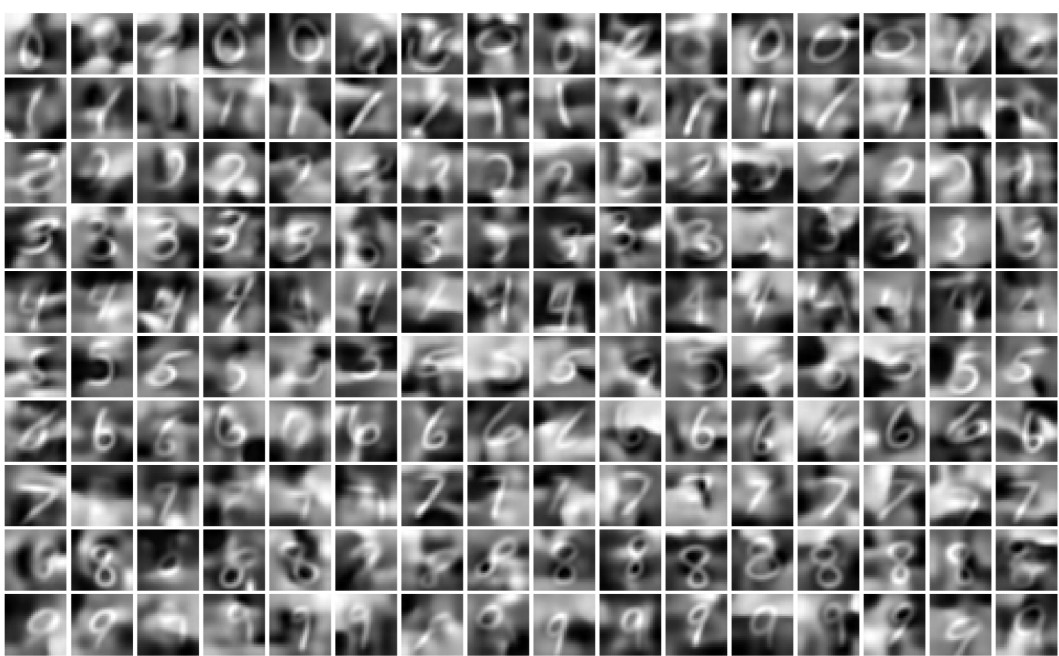

Figure 6: Generative samples from CPVAE trained on Grassy MNIST. Each row is from a population corresponding to a different digit.

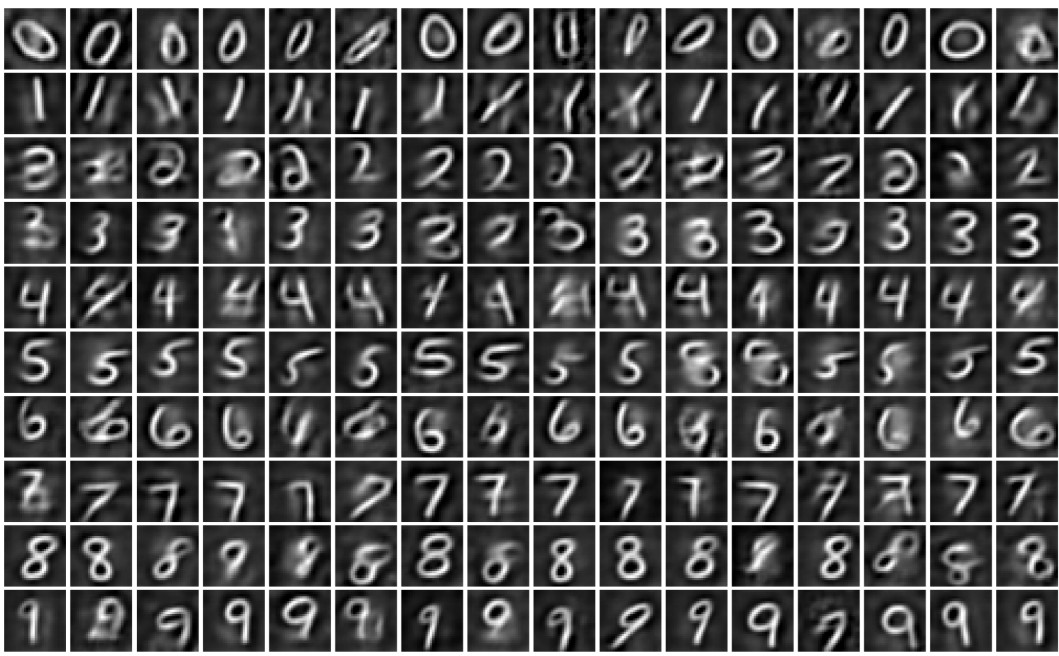

Figure 7: Generative samples from the private space of CPVAE trained on Grassy MNIST. Each row is from a population corresponding to a different digit.

# B  EXPERIMENT DETAILS

## B.1  MAXIMUM MARGINAL LIKLIHOOD CLASSIFICATION

The weight on the mutual information term is annealed according to $\alpha(e) = \min(1.1^e, 1000)$ where $e$ is the training epoch number. For MNIST, the shared and private latent dimensions are 2 and 5, respectively. For CIFAR-10, the latent dimensions are 10 and 20. For Grassy MNIST, they are 30 and 5. We augment each set by randomly translating the images by up to $10\%$ in each direction during training which we find to help reduce overfitting and increase test performance.

## B.2  CONTINUAL LEARNING

The continual learning experiment uses a total of $K = 5$ populations, one for each task. The dimensions of the latent vectors are 10 for the shared space and 10 for each private space. The VAE which we use for comparison has a latent size of 60 so that the total latent size of each model is the same. During the first training phase we use a base learning rate of 0.001, and lower it to 0.0001 during tasks 3-5.

