# OpenReview forum: "Isolating Latent Structure with Cross-population Variational Autoencoders"
_ICLR.cc/2020/Conference — Reject_

### Official Review · AnonReviewer1 · 2019-10-21
**Official Blind Review #1**

**Rating:** 6

**Review:**

This paper studied the problem of learning the latent representation from a complex data set which followed the independent but not identically distributions. The main contributions of this paper are to explicitly learn the commonly shared and private latent factors for different data populations in a unified VAE framework, and propose a mutual information regularized inference in order to avoid the “leaking” induced by the shared representations across different populations. The isolation of the commonly shared and population specific latent representations learned by the proposed are empirically demonstrated on several applications. However, I have some concerns regarding this paper as follows.
(1) It is not clear why the private representation exhibits latent features from the shared space when using equation 3 and how this phenomenon hurts this CPVAE model.
(2) In equation 1, how to define the isotropic diagonal covariance matrix in the Gaussian distribution p? Is it parameterized by g?
(3) In equation 3, what is the prior distribution of p(z_ki, t_ki)?
(4) In equation (4)(5), why could the marginal KL term be canceled out when using I_q(x_k; t_k) - I_q(x_-k; \tilde{t}_k)?
(5) The mutual information regularized inference involved the KL term between any two private factors from different populations. It might be not efficient for optimization. Thus, it will be helpful if the authors provide the model efficiency analysis compared with other baseline methods.

Minor comments:
(1) what is the symbol “n_k”? Did it denote the number of examples for the k-th population?
(2) For mutual information regularized inference, it used two different notations: “I_q(x_k; t_k) - I_q(x_-k; t_k)” and “I_q(x^k; t^k) - I_q(x^-k; t^k)”.


**Experience Assessment:**

I have read many papers in this area.

**Review Assessment: Checking Correctness Of Derivations And Theory:**

I assessed the sensibility of the derivations and theory.

**Review Assessment: Checking Correctness Of Experiments:**

I assessed the sensibility of the experiments.

**Review Assessment: Thoroughness In Paper Reading:**

I read the paper thoroughly.

---

### Official Review · AnonReviewer3 · 2019-10-23
**Official Blind Review #3**

**Rating:** 3

**Review:**

This paper introduces a novel model called "Cross-Population Variational Autoencoder (CPVAE), which is designed to model data from different distributions sharing some common structure. The proposed generative model utilizes both shared and private per-population latent variables. In order to restrict shared latent variables from "leaking" into private representations, the authors introduce an information-theoretic regularizer. This regularizer forces private population representations to: (a) maximize mutual information with input samples from their population, and (b) minimize mutual information with input samples from other populations. In other words, private representations are forced to be "meaningful" on the corresponding population alone.

Quality:
The paper is well-written. I find the proposed method to be quite interesting. The derivations appear to be correct except for possibly the cancellation in Eq. 5, which I originally missed.

Significance:
In my opinion, if sound, the approach discussed in this paper may lead to interesting practical applications and may inspire other methods based on similar ideas.

Originality:
Even though, as authors point out, there is a substantial amount of work in this field, I believe that their approach is novel and has its own merits.

Clarity:
The paper is well written and the material is presented with clarity. In my opinion, the only exception is Section 4.4, which could definitely benefit from a few additional sentences describing the training procedure in more detail. Right now I find it a bit confusing. It would appear that the shared encoder / decoder continue to be trained as new populations arrive. Would this mean that catastrophic forgetting can actually impact this shared representation? And if it changes by a sufficient degree, can it reduce the quality of the generated samples for older populations? If so, I think these points should be mentioned in the text.

Questions and suggestions:
Experiments described in the paper are sufficiently convincing, but there are a few questions that could potentially be better clarified in the paper.

1. After reviewing the text again and seeing the comment of Reviewer #1, I am also confused about the cancellation in  I_q(x_k; t_k) - I_q(x_{-k}; \tilde{t}_k). Is it not true that marginal distributions q_\phi(t) and q_\phi(\tilde{t}) in the KL divergence term in Eq. 5 are different? Unfortunately, the final optimization objective relies on the cancellation of these terms and if they do not cancel, the approach may not be theoretically justified despite producing interesting and compelling results. (This affected the final rating. I will be able to change the rating once this point is clarified.)

2. Another issue is related to the special case when there are several very similar populations. Consider, for example, a case when there are two nearly-identical populations out of many. Using very similar latent variables for two similar populations would be penalized by the regularizer (not too significantly though). I assume that depending on the embedding sizes and the value of alpha (which authors introduce in Section 2.3), the model would either choose to use shared latent variables to encode these populations, or would allow for two nearly-identical private latent representations to exist. I think this is a conceptually important special case that could be mentioned and possibly explained in the paper.

3. I think the paper would also benefit from a clarification regarding the "mixing" function g. Choosing this function to be a simple sum of arguments seems restricting and may be insufficient for some datasets. It does not appear to be the case, but are there any restrictions on g? Can it come from a parametrized function family with parameters being optimized during training?

4. I think the paper would benefit from a more detailed discussion in Section 4.4 (see above).

**Experience Assessment:**

I have published one or two papers in this area.

**Review Assessment: Checking Correctness Of Derivations And Theory:**

I assessed the sensibility of the derivations and theory.

**Review Assessment: Checking Correctness Of Experiments:**

I assessed the sensibility of the experiments.

**Review Assessment: Thoroughness In Paper Reading:**

I read the paper at least twice and used my best judgement in assessing the paper.

---

### Official Review · AnonReviewer2 · 2019-10-28
**Official Blind Review #2**

**Rating:** 3

**Review:**

The paper proposes to model multiple datasets from differing distributions with shared latent structure and private latent factors. The main techniques include architecture design which encourages the isolation of shared and private latent factors and a mutual information-based regularizer. The paper is clearly written and easy to follow. I enjoyed reading it. The experiments support the claim of learned population-specific representations.

However, I found that the paper has some weaknesses:
1. The novelty is not enough. All the techniques involved in the paper are not new but from existing literature. The idea is not new. The authors also mentioned several previous works in Section 3, e.g. Multi-level  VAEs, oi-VAEs.

2. More importantly, I did not see any baselines in the experiments except vanilla VAE. As far as I understand, previous methods can be easily adapted to these tasks. For example, [1] tried continual generative modeling for a sequence of distinct distributions. Many important baselines are missing in the experiments, which makes it hard for me to evaluate how significant the work is.

3. What if the populations are not exclusive? The regularizer enforces them to be isolated but they are not in fact.

4. How did you choose the annealing schedule of $\alpha$ in Section B.1?

Minor:

page 2  eq (1) z_i -> z_{ki}

page 3 last paragraph “it may desirable”

References:

[1] https://arxiv.org/pdf/1705.08395.pdf



**Experience Assessment:**

I have published one or two papers in this area.

**Review Assessment: Checking Correctness Of Derivations And Theory:**

I carefully checked the derivations and theory.

**Review Assessment: Checking Correctness Of Experiments:**

I carefully checked the experiments.

**Review Assessment: Thoroughness In Paper Reading:**

I read the paper thoroughly.

---

### Decision · Program_Chairs · 2019-12-19

**Decision:**

Reject

**Comment:**

The paper proposes a hierarchical Bayesian model over multiple data sets that
has both data set specific as well as shared parameters.
The data set specific parameters are further encouraged to only capture aspects
that vary across data sets by an addition mutual information contribution to the
training loss.
The proposed method is compared to standard VAEs on multiple data sets.

The reviewers agree that the main approach of the paper is sensible. However,
concerns were raised about general novelty, about the theoretical justification
for the proposed loss function and about the lack of non-trivial baselines.
The authors' rebuttal did not manage to full address these points.

Based on the reviews and my own reading, I think this paper is slightly
below acceptance threshold.